# SageMix: Saliency-Guided Mixup for Point Clouds

**Sanghyeok Lee**[*]
Korea University
cat0626@korea.ac.kr

**Minkyu Jeon**[*]
Korea University
jmk94810@korea.ac.kr

**Injae Kim**
Korea University
dna9041@korea.ac.kr

**Yunyang Xiong**
Meta Reality Labs
yunyang@fb.com

**Hyunwoo J. Kim**[†]
Korea University
hyunwoojkim@korea.ac.kr

## Abstract

Data augmentation is key to improving the generalization ability of deep learning models. Mixup is a simple and widely-used data augmentation technique that has proven effective in alleviating the problems of overfitting and data scarcity. Also, recent studies of saliency-aware Mixup in the image domain show that preserving discriminative parts is beneficial to improving the generalization performance. However, these Mixup-based data augmentations are underexplored in 3D vision, especially in point clouds. In this paper, we propose **SageMix**, a saliency-guided Mixup for point clouds to preserve salient local structures. Specifically, we extract salient regions from two point clouds and smoothly combine them into one continuous shape. With a simple sequential sampling by re-weighted saliency scores, SageMix preserves the local structure of salient regions. Extensive experiments demonstrate that the proposed method consistently outperforms existing Mixup methods in various benchmark point cloud datasets. With PointNet++, our method achieves an accuracy gain of 2.6% and 4.0% over standard training in 3D Warehouse dataset (MN40) and ScanObjectNN, respectively. In addition to generalization performance, SageMix improves robustness and uncertainty calibration. Moreover, when adopting our method to various tasks including part segmentation and standard 2D image classification, our method achieves competitive performance. Code is available at https://github.com/mlvlab/SageMix.

## 1 Introduction

Deep neural networks have achieved high performance in various domains including image, video, and speech. Recent researches [1, 2, 3, 4, 5, 6] have been proposed to employ deep learning model for 3D vision, especially in point clouds. However, in the point cloud domain, deep learning models are prone to suffer from weak-generalization performance due to the limited availability of data compared to the image datasets, which contain almost millions of training samples. For alleviating the data scarcity issue, data augmentation is a prevalent solution to increase the training data.

In the image domain, Mixup-based methods [7, 8, 9] combine two training images to generate augmented samples. More recent Mixup methods [10, 11, 12, 13] focus on leveraging saliency to preserve the discriminative regions such as foreground objects. Despite the success of saliency-aware Mixup methods, it has been less studied in point clouds due to its unordered and non-grid structure. Moreover, saliency-aware augmentation methods require solving an additional optimization problem [11, 12], resulting in a considerable computational burden.

---

[*]First two authors have equal contribution.
[†]Corresponding author.

36th Conference on Neural Information Processing Systems (NeurIPS 2022).

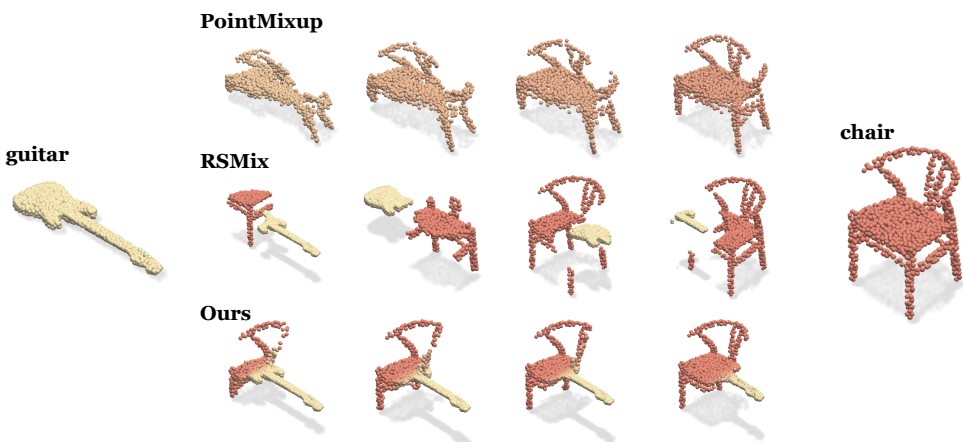

Figure 1: **Comparison of Mixup methods on point clouds.** Given two original samples, (left) guitar and (right) chair, we generate samples by various Mixup methods. (Top) PointMixup does not preserve the discriminative structure. (Middle) The samples generated by RSMix contain the local structure of each sample, but the discontinuity occurs at the border. (Bottom) Our method, SageMix, generates a continuous mixture preserving the local structure of original shapes.

Recently, several works [14, 15, 16] have attempted to extend the concept of Mixup to point clouds. PointMixup [14] enables the linear interpolation of point clouds based on optimal assignment. RSMix [15] proposed a shape-preserving Mixup framework that extracts and merges the rigid subsets of each sample. However, these approaches have limitations. PointMixup generates samples without preserving the local structure of the original shapes. For example, in the top row of Figure 1, the structure of the guitar is not preserved in generated samples. RSmix generates discontinuous samples and these artifacts often hinder effective training. Furthermore, these methods disregard the saliency, thereby causing the loss of the discriminative local structure in the original point cloud.

In this paper, we propose a **Sa**liency-**G**uid**e**d **Mix**up for point clouds (**SageMix**) that preserves discriminative local structures and generates continuous samples with smoothly varying mixing ratios. For saliency estimation, we measure the contribution of each point by the gradients of a loss function. Through a simple sequential sampling via re-weighted saliency scores, SageMix samples the query points to extract salient regions without solving additional optimization problems. Then, SageMix smoothly combines point clouds, considering the distance to query points to minimize the loss of discriminative local structures as illustrated in Figure 1 (*e.g.,* the neck of a guitar and the back of a chair, etc.).

In summary, our **contributions** are fourfold:

- We propose a novel saliency-guided Mixup method for point clouds. To the best of our knowledge, this is the first work that utilizes saliency for Mixup in point clouds.

- We design a Mixup framework that preserves the salient local structure of original shapes while smoothly combining them into one continuous shape.

- Extensive experiments demonstrate that SageMix brings consistent and significant improvements over state-of-the-art Mixup methods in generalization performance, robustness, and uncertainty calibration.

- We demonstrate that the proposed method is extensible to various tasks including part segmentation and standard image classification.

## 2 Preliminaries

In this section, we briefly review the basic concept of Mixup and summarize the variants of Mixup for images and point clouds (Table 1).

Table 1: **Variants of Mixup.**

| Method | | Equation |
|---|---|---|
| Image | mixup [9] | $\tilde{x} = \lambda x_\alpha + (1 - \lambda) x_\beta$ |
| | Manifold mixup [7] | $\tilde{h} = \lambda h_\alpha + (1 - \lambda) h_\beta$ |
| | CutMix [8] | $\tilde{x} = M \odot x_\alpha + (1 - M) \odot x_\beta$ |
| | SaliencyMix [13] | $\tilde{x} = M(x_\alpha) \odot x_\alpha + (1 - M(x_\alpha)) \odot x_\beta$ |
| | Puzzle Mix [11] | $\tilde{x} = Z \odot \Pi_\alpha^\top x_\alpha + (1 - Z) \odot \Pi_\beta^\top x_\beta$ |
| | Co-Mixup [12] | $\tilde{x} = \sum_i^m Z_i \odot x_i$ |
| Point cloud | PointMixup [14] | $\tilde{\mathcal{P}} = \{\lambda p_i^\alpha + (1 - \lambda) p_{\phi(i)}^\beta\}_i^n$ |
| | RSMix [15] | $\tilde{\mathcal{P}} = (\mathcal{P}^\alpha - \mathcal{S}^\alpha) \cup \mathcal{S}^{\beta \to \alpha}$ |
| | **SageMix** | $\tilde{\mathcal{P}} = \{\lambda_i p_i^\alpha + (1 - \lambda_i) p_{\phi(i)}^\beta\}_i^n$ |

**Vicinal risk minimization with Mixup.** Given observed data $\mathcal{D} = \{(x_i, y_i)\}_i^m$ and a function $f : \mathcal{X} \to \mathcal{Y}$, Chapelle et al. [17] learns the function $f$ by minimizing the empirical vicinal risk: $R_\nu(f) = \frac{1}{m'} \sum_{i=1}^{m'} \ell(f(\tilde{x}_i), \tilde{y}_i)$, where $\ell$ is the loss function, and $(\tilde{x}, \tilde{y})$ is the virtual feature-target pair from the vicinal distribution $\nu$ of the observed data $\mathcal{D}$. To construct an effective vicinal distribution in the image domain, Zhang et al. [9] introduced Mixup:

$$\tilde{x} = \lambda x_\alpha + (1 - \lambda) x_\beta, \quad \tilde{y} = \lambda y_\alpha + (1 - \lambda) y_\beta, \tag{1}$$

where $(x_\alpha, y_\alpha), (x_\beta, y_\beta)$ are two pairs of data in training dataset $\mathcal{D}$, and $\lambda \sim \text{Beta}(\theta, \theta)$ is a mixture ratio. Following [9], Verma et al. [7] proposed Manifold Mixup that applies Mixup in hidden representations (*i.e.,* $\tilde{h} = \lambda h_\alpha + (1 - \lambda) h_\beta$) and CutMix [8] generates samples via cut-and-paste manner with binary mask $M \in \{0, 1\}^{W \times H}$ (*i.e.,* $\tilde{x} = M \odot x_\alpha + (1 - M) \odot x_\beta$, where $\odot$ indicates the element-wise product). Combining with saliency, SaliencyMix [13] improved CutMix by selecting the patch $M(x_\alpha)$ with the maximum saliency values. Puzzle Mix optimizes the mask $Z \in [0, 1]^{W \times H}$ and transport $\Pi$ for maximizing the saliency of the mixed sample (*i.e.,* $\tilde{x} = Z \odot \Pi_\alpha^\top x_\alpha + (1 - Z) \odot \Pi_\beta^\top x_\beta$). Beyond two samples, Co-Mixup mixes multiple samples in a mini-batch by optimizing multiple masks (*i.e.,* $\tilde{x} = \sum_i^m Z_i \odot x_i$).

While saliency-aware Mixup methods boost the generalization of deep learning models, they are originally designed for the image domain. Hence, these methods are not directly applicable to point clouds due to the unordered and irregular structure.

**Mixup in point cloud.** Several works [14, 15] tried to leverage the Mixup in point cloud. Point-Mixup [14] linearly interpolates two point clouds by

$$\tilde{\mathcal{P}} = \{\lambda p_i^\alpha + (1 - \lambda) p_{\phi^*(i)}^\beta\}_i^n, \quad \tilde{y} = \lambda y^\alpha + (1 - \lambda) y^\beta, \tag{2}$$

$$\phi^* = \underset{\phi \in \Phi}{\operatorname{argmin}} \sum_i^n \|p_i^\alpha - p_{\phi(i)}^\beta\|_2, \tag{3}$$

where $\mathcal{P}^t = \{p_1^t, ..., p_n^t\}$ is the set of points with $t \in \{\alpha, \beta\}$, $n$ is the number of points, and $\phi^* : \{1, ..., n\} \mapsto \{1, ..., n\}$ is the optimal bijective assignment between two point clouds. In RSMix [15], they generate an augmented sample by merging the subsets of two objects, defined as $\tilde{\mathcal{P}} = (\mathcal{P}^\alpha - \mathcal{S}^\alpha) \cup \mathcal{S}^{\beta \to \alpha}$, where $\mathcal{S}^t \subset \mathcal{P}^t$ is the rigid subset and $\mathcal{S}^{\beta \to \alpha}$ denotes $S^\beta$ translated to the center of $\mathcal{S}^\alpha$.

Although these methods have shown that Mixup is effective for point clouds, some limitations have remained unresolved: loss of original structures, discontinuity at the boundary, and loss of discriminative regions. Here, to address these issues, we propose a new Mixup framework in the following section.

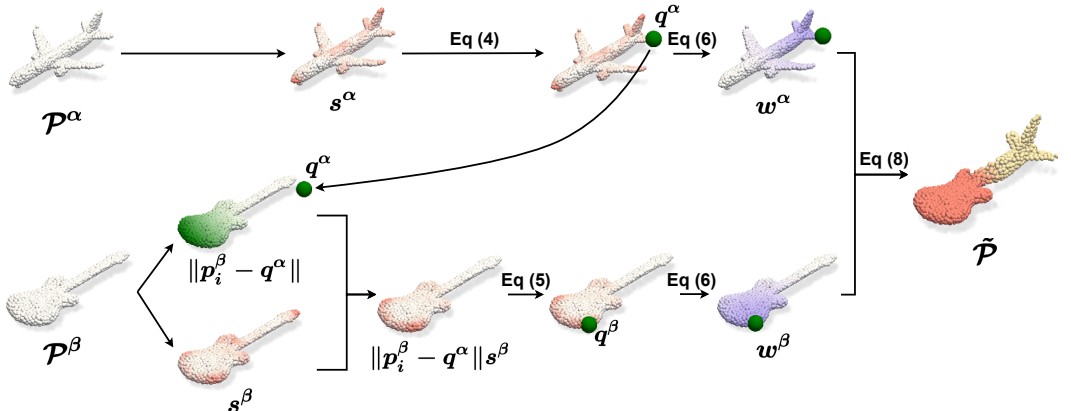

Figure 2: **Illustration of SageMix pipeline.** Given a pair of samples $\mathcal{P}^\alpha$, and $\mathcal{P}^\beta$, SageMix sequentially samples the query points $q^\alpha$, and $q^\beta$ (green) based on saliency using Equation (4) and Equation (5), respectively. Then, using smoothly varying weights with respect to the distance to query points, SageMix generates an augmented sample $\tilde{\mathcal{P}}$ preserving the salient local structures.

## 3 Method

The main goal of SageMix is to generate an augmented sample that 1) maximally preserves the salient parts, 2) keeps the local structure of original shapes, and 3) maintains the continuity in the boundary. To achieve this goal, SageMix extracts salient regions centered at query points and smoothly combines them through continuous weights in Euclidean space. The overall pipeline is illustrated in Figure 2 and pseudocode is provided in Algorithm 1 . In this section, we delineate the process for selecting the query point based on the saliency map in Section 3.1 and Mixup methods for preserving the salient local structure in Section 3.2.

### 3.1 Saliency-guided sequential sampling

We first introduce a query point $q^t \in \mathcal{P}^t$ that is considered the center of the region for preserving the local shape. A naïve random sampling for a query point is simple, but it does not guarantee that it is placed in a salient region. Thus, we propose saliency-guided query point selection for maximally maintaining the salient part. We denote the saliency $S^t$ of the input $\mathcal{P}^t$ by the norm of the gradient (*i.e.,* $S^t = \|\nabla_{\mathcal{P}^t}\ell(f(\mathcal{P}^t), y^t)\|$), following [11, 12]. Deterministically selecting a query point with a maximum saliency score (*i.e.,* $q^t = p_{i*}^t$, where $i^* = \arg\max_i(S_i^t)$) sounds promising but in practice, this makes it difficult to generate diverse samples since query points are always located at the same position. Moreover, if two query points $q^\alpha, q^\beta$ are closely located in the Euclidean space, it is challenging to preserve the local structure of each sample because of the significant overlap.

To address these issues, SageMix sequentially selects the query point based on the saliency scores. The first step is to extract the query point $q^\alpha = p_{i*}^\alpha$. To maximize the diversity, the query point is sampled with respect to the probability distribution defined as

$$\Pr(I^\alpha = i) = \frac{s_i^\alpha}{\sum_i^n s_i^\alpha}, \tag{4}$$

where $s^\alpha \in S^\alpha$ is the saliency of $p_i^\alpha$ and $I^\alpha$ is a random variable for index ($i^* \sim I^\alpha$). That is, the points in a salient region have a high chance to be chosen as a query point. This simple sampling method efficiently provides diverse query points, thereby minimizing redundant selections. Given query point $q^\alpha$, the next step is to define a sampling method for $q^\beta$ to alleviate the overlap between selected parts. We encourage the sampler to distance $q^\beta$ from $q^\alpha$ by reweighting the saliency scores:

$$\Pr(I^\beta = i) = \frac{\|p_i^\beta - q^\alpha\|s_i^\beta}{\sum_i^n \|p_i^\beta - q^\alpha\|s_i^\beta}. \tag{5}$$

---

**Algorithm 1 A saliency-guided Mixup for point clouds**

---

**Input:** $\mathcal{P}^\alpha, \mathcal{P}^\beta, y^\alpha, y^\beta, S^\alpha, S^\beta, \sigma, \phi, \theta$
$\mathcal{P} = \{p_1, ..., p_n\}$ : set of points, $y$ : target, $S = \{s_1, ..., s_n\}$ : saliency values, $\sigma$ : bandwidth, $\phi$ : assignment function, $\theta$ : shape parameter
**Output:** $\tilde{\mathcal{P}}, \tilde{y}$

1: Draw $q^\alpha = p_i^\alpha$ from $\mathcal{P}^\alpha$ w.r.t. $\Pr(I^\alpha = i) = \frac{s_i^\alpha}{\sum_i^n s_i^\alpha}$        $\triangleright$ Equation (4)

2: Draw $q^\beta = p_i^\beta$ from $\mathcal{P}^\beta$ w.r.t. $\Pr(I^\beta = i) = \frac{\|p_i^\beta - q^\alpha\| s_i^\beta}{\sum_i^n \|p_i^\beta - q^\alpha\| s_i^\beta}$    $\triangleright$ Equation (5)

3: **for** $i = 1$ to $n$ **do**

4:   $w_i^\alpha, w_i^\beta \leftarrow K_\sigma(p_i^\alpha, q^\alpha), K_\sigma(p_i^\beta, q^\beta)$         $\triangleright$ Equation (6)

5:   $\lambda_i \leftarrow \frac{\pi w_i^\alpha}{\pi w_i^\alpha + (1-\pi) w_{\phi(i)}^\beta}$      $\triangleright \pi \sim \text{Beta}(\theta, \theta)$, Equation (9)

6:   $\tilde{p}_i \leftarrow \lambda_i p_i^\alpha + (1 - \lambda_i) p_{\phi(i)}^\beta$          $\triangleright$ Equation (8)

7: **end for**

8: $\lambda \leftarrow \frac{1}{n} \Sigma_i^n \lambda_i$

9: $\tilde{y} \leftarrow \lambda y^\alpha + (1 - \lambda) y^\beta$

10: **return** $\tilde{\mathcal{P}} = \{\tilde{p}_1, .., \tilde{p}_n\}, \tilde{y}$

---

Assuming the point $p_i^\beta \to q^\alpha$, the probability of selecting the point $p_i^\beta$ decreases. This implies that SageMix samples a query point considering both distance and saliency. Finally, with Equation (4) and Equation (5), we can obtain the remotely located query points with high saliency scores, resulting in augmented samples preserving the discriminative structures of original shapes.

## 3.2 Shape-preserving continuous Mixup

Since our objective is to preserve the region around the query point $q^t$, we need to impose high weight on the points near the query point. Further, to alleviate the discontinuity between two samples, the weight should smoothly vary in Euclidean space. Herein, we use a Gaussian Radial Basis Function (RBF) kernel to calculate weights:

$$w_i^t = K_\sigma(p_i^t, q^t) = \exp\left(-\frac{\|p_i^t - q^t\|^2}{2\sigma^2}\right), \quad (6)$$

where $w_i^t$ is the weight on the point $p_i^t$ and $\sigma \in \mathbb{R}_+$ is a bandwidth for kernel. The weight $w_i^t$ smoothly increases as the distance to $q^t$ decreases, resulting in the region around the query point being prone to respect its original shape. As shown in (Figure 2), the parts with higher weights maintain their original local structure more in a mixed sample, *e.g.,* the body of the guitar and the tail of the airplane.

Given two point clouds $(\mathcal{P}^\alpha, \mathcal{P}^\beta)$ and their corresponding weights $(\{w_i^\alpha\}_i^n, \{w_i^\beta\}_i^n)$, we generate an augmented sample via point-wise interpolation. For differentially mixing points, we define the mixing ratio for the $i$-th point pair as

$$\lambda_i^\alpha = \frac{w_i^\alpha}{w_i^\alpha + w_{\phi(i)}^\beta}, \quad \lambda_{\phi(i)}^\beta = \frac{w_{\phi(i)}^\beta}{w_i^\alpha + w_{\phi(i)}^\beta}, \quad (7)$$

where $\phi$ is the assignment in Equation (3). Note, that the point-wise mixing ratio $\lambda_i^\alpha$ is the ratio between the weight of two paired points (*i.e.,* $\lambda_{\phi(i)}^\beta = 1 - \lambda_i^\alpha$), thus enabling linear interpolation. For simplicity, we use the notation $\lambda_i, (1 - \lambda_i)$ instead of $\lambda_i^\alpha, \lambda_{\phi(i)}^\beta$. Then, we generate the virtual point cloud by modifying the Equation (2) to

$$\tilde{\mathcal{P}} = \{\lambda_i p_i^\alpha + (1 - \lambda_i) p_{\phi(i)}^\beta\}_i^n, \quad \tilde{y} = \lambda y^\alpha + (1 - \lambda) y^\beta, \quad (8)$$

where $\lambda = \frac{1}{n} \sum_i^n \lambda_i$ can be interpreted as the overall mixing ratio that is used in label interpolation. In Mixup, the distribution of $\lambda$ is a key factor for model training [9]. To control the distribution of $\lambda$,

Table 2: **3D shape classification performance on MN40/OBJ_ONLY/PB_T50_RS.**

| Model | Method | Dataset | | |
|---|---|---|---|---|
| | | MN40 | OBJ_ONLY | PB_T50_RS |
| PointNet [3] | Base | 89.2 | 79.1 | 65.4 |
| | + PointMixup [14] | 89.9 | 79.4 | 65.7 |
| | + RSMix [15] | 88.7 | 77.8 | 65.7 |
| | + SageMix | **90.3** | **79.5** | **66.1** |
| PointNet++ [4] | Base | 90.7 | 86.5 | 79.7 |
| | + PointMixup [14] | 92.3 | 87.6 | 80.2 |
| | + RSMix [15] | 91.6 | 87.4 | 81.1 |
| | + SageMix | **93.3** | **88.7** | **83.7** |
| DGCNN [6] | Base | 92.9 | 86.2 | 79.9 |
| | + PointMixup [14] | 92.9 | 86.9 | 82.5 |
| | + RSMix [15] | 93.5 | 86.6 | 82.2 |
| | + SageMix | **93.6** | **88.0** | **83.6** |

we introduce a prior factor $\pi \sim \text{Beta}(\theta, \theta)$:

$$\lambda_i = \frac{\pi w_i^\alpha}{\pi w_i^\alpha + (1 - \pi) w_{\phi(i)}^\beta}. \tag{9}$$

*Remarks.* Our SageMix can emulate both PointMixup and RSMix depending on the bandwidth $\sigma$. If the bandwidth $\sigma$ is sufficiently large, $K_\sigma(p_i^t, q^t) \approx K_\sigma(p_j^t, q^t) = c, \forall_{i \neq j}$, where $c \in \mathbb{R}_+$, resulting in $\lambda_i \approx \frac{\pi c}{\pi c + (1 - \pi) c} = \pi$. That is, all points are mixed in the same ratio as PointMixup. Conversely, when the bandwidth becomes smaller, SageMix changes the mixing ratio drastically around the boundary between the two shapes and generates an augmented sample like RSMix. Qualitative results are available in Section 4.2. See Section 4.2 for qualitative results. Additionally, with a minor change, SageMix is applicable in feature space as Manifold Mixup, see Appendix B.2 for more discussion.

## 4 Experiments

In this section, we demonstrate the effectiveness of our proposed method SageMix with various benchmark datasets. First, for 3D shape classification, we evaluate the generalization performance, robustness, and calibration error in Section 4.1. Next, we provide an ablation study and analyses of SageMix in Section 4.2. Lastly, we study the extensibility of our method in Section 4.3. Implementation details are provided in Appendix A.

**Data.** We use two benchmark dataset: 3D Warehouse dataset (**MN40**) [18] and ScanObjectNN [19]. MN40 is a synthetic dataset containing 9,843 CAD models for training and 2,468 CAD models for evaluation. Each CAD model of MN40 is obtained from 3D Warehouse [20]. ScanObjectNN, obtained from SceneNN [21] and ScanNet[22], is a real-world dataset that is split into 80% for training and 20% for evaluation. Among the variants of ScanObjectNN, we adopt the simplest version (**OBJ_ONLY**) and the most challenging version (**PB_T50_RS**). For training models, we use only coordinates (x,y,z) of 1024 points without additional information such as the normal vector.

**Baselines.** For a comparison with previous studies, we use three backbone models: **PointNet** [3], **PointNet++** [4], and **DGCNN** [6]. We compare SageMix with the model under default augmentation in [3, 4, 6, 19] (**Base**), and other Mixup approaches (**PointMixup** [14], **RSMix** [15]). We report

Table 3: **Robustness and calibration with DGCNN on OBJ_ONLY.**

| Method | Gaussian noise. | | Rotation 180° | | Scaling | | Dropout | | Calibration |
|---|---|---|---|---|---|---|---|---|---|
| | $\sigma' : 0.01$ | $\sigma' : 0.05$ | X-axis | Z-axis | ×0.6 | ×2.0 | 25% | 50% | ECE(%) |
| DGCNN [6] | 84.9 | 48.4 | 32.5 | 32.4 | 73.7 | 73.0 | 83.3 | 75.7 | 19.8 |
| + PointMixup [14] | 85.0 | **61.3** | 31.7 | 32.7 | 73.8 | 73.0 | 84.2 | 74.9 | 6.8 |
| + RSMix [15] | 84.2 | 49.1 | 32.7 | 32.6 | 75.0 | 74.5 | 84.0 | 73.6 | 18.9 |
| + **SageMix** | **85.7** | 51.2 | **36.5** | **37.9** | **75.6** | **75.2** | **84.9** | **79.0** | **5.1** |

Table 4: **Ablation of Saliency-guided sequential sampling.**

| Metric | DGCNN [6] | Uniform | Max | Saliency only | **SageMix** |
|---|---|---|---|---|---|
| OA | 86.2 | 86.8 | 86.1 | 87.8 | **88.0** |

the performance in overall accuracy. We highlight the best performance in red and second-best performance in yellow.

## 4.1  3D shape classification.

**Generalization performance.**  Table 2 summarizes the experimental results of 3D shape classification on three datasets. Our framework significantly outperforms all of the previous methods in every dataset and model. Although the datasets are quite saturated, the averages of the improvements against the Base with PointNet, PointNet++, and DGCNN are 0.7%, 2.9%, and 2.1%, respectively. With PointNet++, SageMix improves the overall accuracy by 2.6%, 2.2%, and 4.0% compared to Base in MN40, OBJ_ONLY, and PB_T50_RS, respectively. We observe similar performance improvements in DGCNN by 0.7%, 1.8%, and 3.7% over Base. These consistent improvements demonstrate the effectiveness of our framework.

**Robustness.**  We adopt DGCNN and OBJ_ONLY to evaluate the robustness of models trained by our method. We compare our method with previous methods [14, 15] on four types of corruption: (1) jittering the point cloud with **Gaussian noise** ($\sigma' \in 0.01, 0.05$), (2) **Rotation 180°**, (3) **Scaling** with a factor in {0.6, 2.0}, and (4) **Dropout** 25% or 50% of all points. As shown in Table 3, SageMix consistently improves the robustness in various corruption. Except for Gaussian noise with $\sigma' = 0.05$, DGCNN trained with SageMix shows the best robustness with significant gains compared to previous methods. Specifically, SageMix outperforms the other methods with gains of 5.5% for rotation (Z-axis), 2.2% for scaling (×2.0), and 3.3% for dropout (50%) over base DGCNN.

**Calibration.**  Previous works [23, 24] have proven that deep neural networks tend to be over-confident resulting in poorly calibrated models. In other words, a well-calibrated model should provide an accurate probability according to its predictions. Here, we report *Expected Calibration Error* (ECE) [23] to estimate the uncertainty calibration of models. We use the same setting as the robustness test for reporting ECE (Table 3). Overall, this result reveals that SageMix provides the best performance on uncertainty calibration compared to the other methods. See Appendix B.3 for more results.

## 4.2  Ablation study and analyses

We provide various quantitative and qualitative analyses for a better understanding of SageMix. We use DGCNN and OBJ_ONLY for an ablation study and MN40 for visualization.

**Ablation on query point sampling.**  We explore the effectiveness of saliency-guided sequential query point sampling. Table 4 shows the results with various sampling methods for query points. A naïve uniform sampling for query point (**Uniform**), without considering the saliency, introduces +0.6% gains over base DGCNN. Interestingly, when SageMix (deterministically) selects the query point with maximum saliency values (**Max**), the performance is even degraded (-0.1%). With our

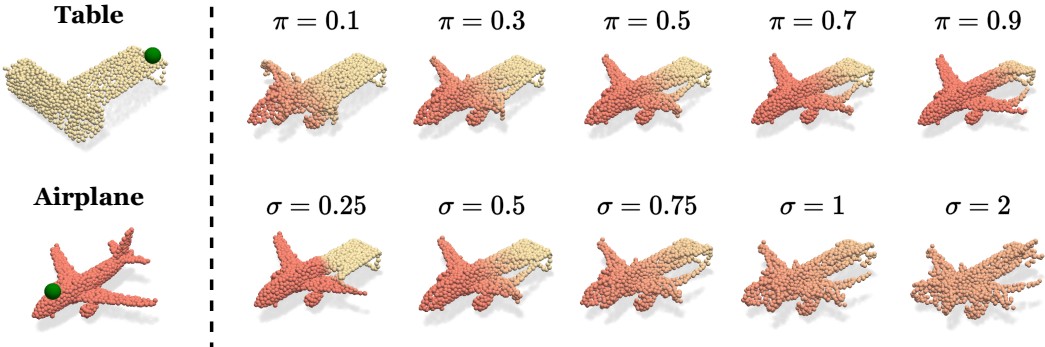

Figure 3: **Qualitative analyses on prior factor $\pi$ and the bandwidth $\sigma$.** Given two samples (left) table and airplane, SageMix generates a sample based on (Top) various $\pi$ with a fixed $\sigma = 0.3$ and (Bottom) various $\sigma$ with a fixed $\pi = 0.5$. Note that the green points indicate query points.

Table 5: **Quantitative analysis on the bandwidth $\sigma$**

| $\sigma$ | 0.1 | 0.3 | 0.5 | 1.0 | 2.0 |
|---|---|---|---|---|---|
| OA | 87.2 | **88.0** | 87.6 | 87.3 | 87.6 |

saliency-guided random sampling for two query points based only on Equation (4) (**Saliency Only**), the performance is significantly improved to 87.8% (+1.6%). These results suggest that although saliency is key to effective training, we also have to consider the diversity for generating a new sample. Finally, the best accuracy of 88.0% (+1.8%) is obtained by our saliency-guided sequential sampling considering saliency and the distance between query points together.

**Prior factor.** We introduce the parameter $\pi$ as a prior factor for mixing ratio $\lambda$. The top row of Figure 3 is the visualization of samples generated by SageMix with various prior factor $\pi$. Given two samples and their corresponding query points colored in green, SageMix generates a sample with the salient local regions from both samples, *e.g.,* the head and right-wing of an airplane and the back of a table are preserved. We observe that the generated sample gets close to the airplane as $\pi \to 1$, and vice versa. In short, SageMix controls the distribution of mixing ratio $\lambda$ based on $\pi$.

**Bandwidth.** The bandwidth $\sigma$ of the RBF kernel controls the change of point-wise mixing ratios in SageMix. As mentioned in Section 4.2, when the bandwidth is sufficiently large ($\sigma = 2$), SageMix emulates PointMixup [14]. SageMix tends to yield globally even weights and constant mixing ratios for all points rather than focusing on local parts. In contrast, as the bandwidth gets smaller, SageMix tends to impose higher weights around the query point preserving the local structure more precisely. These are well exemplified in the bottom row of Figure 3. For instance, when $\sigma = 0.25$, we notice the steep change in the boundary of two samples while preserving the salient local structure. This allows generating augmented samples like RSMix. In short, our SageMix can exhibit similar behaviors as PointMixup and RSMix depending on the bandwidth $\sigma$. We also share the quantitative analysis of the bandwidth with DGCNN and OBJ_ONLY in Table 5. We observed that SageMix with a wide range of bandwidth (0.1 to 2.0) consistently outperforms previous Mixup methods (e.g., 86.9%, 86.6% for PointMixup, RSMix).

### 4.3 Extensions to part segmentation and 2D image classification.

**Part segmentation.** For part segmentation, we train DGCNN on 3D Warehouse (**SN-Parts**) [20, 18]. In part segmentation, since a model predicts a label for each point $\tilde{p}_i$, we generate point-wise ground truth, *i.e.,* $\tilde{y}_i = \lambda_i y_i^\alpha + (1 - \lambda_i) y_{\phi(i)}^\beta$. Also, for a comparison with previous methods, we used the official code by the authors of PointMixup and RSMix with minor modifications for generating

Table 6: **Part segmentation performance on SN-Parts [18].**

| Model | Base | PointMixup [14] | RSMix [15] | **SageMix** |
|---|---|---|---|---|
| PointNet++ [4] | 85.1 | 85.5 | 85.4 | **85.7** |
| DGCNN [6] | 85.1 | 85.3 | 85.2 | **85.4** |

Table 7: **2D image classification performance with PreActResNet18 [25] on CIFAR-100.**

| Dataset | Vanilla | Mixup | Manifold | CutMix | SaliencyMix | Puzzle Mix | Co-Mixup | **SageMix** |
|---|---|---|---|---|---|---|---|---|
| CIFAR-100 | 76.41 | 77.57 | 78.36 | 78.71 | 79.06 | 79.38 | 80.13 | **80.16** |

point-wise ground truth. We follow the settings in [4, 6] to evaluate our method and reports the results in Table 6. Note that although the gain seems small, SageMix outperforms previous Mixup methods. Also, considering the already saturated performance, we believe that the improvement (+0.6%, +0.3% in PointNet++, DGCNN) over the base model is noteworthy.

**2D classification.** Our framework is also applicable to 2D image classification. Following [12], we used PreActResNet18 [25] in the CIFAR-100 for our experiments. We compare our method with several Mixup baselines in the image domain [11, 12, 13, 7, 9, 8]. Although our method is designed for point clouds, it shows competitive performance. SageMix achieves the best accuracy of 80.16% with PreActResNet18 on the CIFAR-100 dataset (Table 7). It is worth noting that PuzzleMix and Co-Mixup require an additional optimization which introduces a considerable computational overhead. Specifically, SageMix is $\times 6.05$ faster than Co-Mixup [12] per epoch in the CIFAR-100 dataset. We believe that our simple sampling technique is helpful to improve the generalization power of the model. For more details, see Appendix B.4.

## 5 Related works

**Deep learning on point clouds.** PointNet [3] is a pioneering work that designs a novel deep neural network for processing unordered 3D point sets with a multi-layer perceptron. Inspired by CNNs, Qi et al. [4] propose PointNet++ with a hierarchical architecture. In DGCNN, Wang et al. [6] introduce EdgeConv which utilizes edge features from the dynamically updated graph. Additionally, various works have focused on point-wise multi-layer perceptron [26, 27, 28], convolution [29, 30, 31, 1, 2, 5, 32], and graph-based methods [33, 34] to process point clouds. Parallel to these approaches, other recent works [14, 35, 15, 36, 8] focus on data augmentation to improve the generalization power of deep neural networks in point clouds.

**Mixup.** Mixup [9] is a widely used regularization technique, which linearly interpolates a pair of images to generate an augmented sample. Following this work, Verma et al. [7] propose Manifold Mixup that extends Mixup to the hidden representations. CutMix [8] replaces a part of an image with a part of another one. More recent studies [11, 12, 13] have been proposed to preserve the saliency while mixing samples. In point clouds, PointMixup [14] is the first approach that adapts the concept of Mixup in point clouds with the optimal assignments. Instead of linear interpolation, Lee et al. [15] propose RSMix which merges the subsets of two point clouds inspired by CutMix.

**Saliency.** Measuring the saliency of data using neural networks has been studied to obtain a more precise saliency map [37, 38, 39]. The saliency has been prevalent in various fields such as object detection, segmentation, and speech recognition [10, 40, 41, 42, 43]. Similarly, PointCloud Saliency Map [44] constructed the saliency map to identify the critical points via building a gradient-based saliency map. Recently, saliency has been used in Mixup framework [11, 12, 13] to prevent generating samples only with background or irrelevant regions to the target objects.

# 6   Conclusion

We propose SageMix, a novel saliency-guided Mixup for point clouds to preserve salient local structures. Our method generates an augmented sample with a continuous boundary while preserving the discriminative regions. Additionally, with a simple saliency-guided sequential sampling, SageMix achieves state-of-the-art performance in various metrics (*e.g.,* generalization, robustness, and uncertainty calibration). Moreover, we demonstrate that the proposed method is extensible to various tasks: part segmentation and standard 2D image classification. The visualization supports that SageMix generates a continuous mixture while respecting the salient local structure.

## Acknowledgments and Disclosure of Funding

This work was partly supported by ICT Creative Consilience program (IITP-2022-2020-0-01819) supervised by the IITP; the National Supercomputing Center with supercomputing resources including technical support (KSC-2022-CRE-0261); and IITP grant funded by the Korea government (MSIT) (No.2021-0-02312, Efficient Meta-learning Based Training Method and Multipurpose Multi-modal Artificial Neural Networks for Drone AI).

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
