# Appendix

## A    Implementation details

We conduct experiments using Python and PyTorch[1] [1] with a single NVIDIA TITAN RTX for point clouds and NVIDIA RTX 3090 for 2D image classification. Following the original configuration in [2, 3, 4], we use the Adam [5] optimizer with an initial learning rate of $10^{-3}$ for PointNet[2] [2] and PointNet++[2] [3] and SGD with an initial learning rate of $10^{-1}$ for DGCNN[3] [4]. We train models with a batch size of 32 for 500 epochs. For a fair comparison with previous works [6, 7], we also adopt conventional data augmentations with our framework (*i.e.,* scaling and shifting for MN40 [8] and rotation and jittering for ScanObjectNN [4][9]). When the performance of a baseline on ScanObjectNN is unavailable in the original paper of PointMixup [6] and RSMix[5] [7], we reproduce the results based on their official code. For hyperparameters of SageMix, we opt $\theta = 0.2$ in entire experiments. Regarding the bandwidth for RBF kernel, we opt $\sigma = 2.0$ for PointNet and $\sigma = 0.3$ for PointNet++ and DGCNN.

## B    Additional Experiments

### B.1    Error bars

Performance oscillation is an important issue in point cloud benchmarks. However, for a fair comparison with the numbers reported in PointMixup [6] and RSMix [7], we followed the prevalent evaluation metric in point clouds, which reports the best validation accuracy. Apart from this, we here provide the additional results with five runs on OBJ_ONLY. The mean and standard deviation are presented in Table 1.

Table 1: **Mean and standard deviation measures on OBJ_ONLY.**

| Method | Model | | |
|---|---|---|---|
| | PointNet [2] | PointNet++ [3] | DGCNN [4] |
| Base | 78.56±0.51 | 86.14±0.39 | 85.72±0.44 |
| + PointMixup [6] | 78.88±0.28 | 87.50±0.26 | 86.26±0.34 |
| + RSMix [7] | 77.60±0.56 | 87.30±0.65 | 85.88±0.59 |
| **+ SageMix** | **79.14±0.30** | **88.42±0.26** | **87.32±0.53** |

### B.2    Manifold mixup

We train DGCNN [4] to validate the SageMix in a feature space. Following manifold Mixup [10], we apply SageMix in a randomly selected layer. The results are summarized in Table 2. We observe the competitiveness of SageMix in feature space with the performance improvements by 0.6%, 1.5%, 3.3% in MN40, OBJ_ONLY, and PB_T50_RS, respectively.

### B.3    Uncertainty calibration

In this section, we measure the Expected Calibration Error (ECE) [11] of the model on three datasets. As shown in Table 3, our model consistently has the lowest calibration error on every dataset. Specifically, SageMix lowers ECE by 16.1%, 14.7%, and 15.6% compared to vanilla DGCNN in MN40, OBJ_ONLY, and PB_T50_RS, respectively.

---

[1]©2016 Facebook, Inc (Adam Paszke). Licensed under BSD-3-Clause License

[2]©2017 Charles R. Qi. Licensed under MIT License

[3]©2019 Yue Wang. Licensed under MIT License

[4]©2019 Vision & Graphics Group, HKUST. Licensed under MIT License

[5]©2020 dogyoonlee. Licensed under MIT License

Table 2: **SageMix in input and feature space.**

| Method | MN40 | OBJ_ONLY | PB_T50_RS |
|---|---|---|---|
| DGCNN [4] | 92.9 | 86.2 | 79.9 |
| + **SageMix** (Input Space) | 93.6 | 88.0 | 83.6 |
| + **SageMix** (Feature Space) | 93.5 | 87.7 | 83.2 |

Table 3: **Expected calibration error with DGCNN.**

| Dataset | Vanilla | PointMixup [6] | RSMix [7] | **SageMix** |
|---|---|---|---|---|
| MN40 | 18.3 | 2.4 | 24.2 | **2.2** |
| OBJ_ONLY | 19.8 | 6.8 | 18.9 | **5.1** |
| PB_T50_RS | 18.9 | 4.2 | 16.7 | **3.3** |

## B.4 Detailed results of 2D classification

We largely follow the setting in Co-Mixup[6] [12] except for the learning rate. We trained 300 epochs with the batch size of 128. We adopt SGD as an optimizer with an initial learning rate of 0.1. We set the weight decay and the momentum as $10^{-4}$ and 0.9, respectively. We consider the column number and the row number as the coordinates of each pixel. For SageMix, we use $\theta = 0.3$ and $\sigma = 8$. In Table 4, we report the accuracy and latency for each method. The second row of the table shows the running time per epoch. Our method is $\times 6.05$ faster than Co-Mixup [12]. It is worth noting that our framework achieves state-of-the-art performance with a tolerable computational cost considering the improvements.

Table 4: **2D classification with PreActResNet18 [13] on CIFAR-100.**

| | Vanilla | Mixup | Manifold | CutMix | SaliencyMix | Puzzle Mix | Co-Mixup | Ours |
|---|---|---|---|---|---|---|---|---|
| ACC. (%) | 76.41 | 77.57 | 78.36 | 78.71 | 79.06 | 79.38 | 80.13 | **80.16** |
| Time.(sec) | 13.1 | 20.4 | 20.8 | 23.4 | 21.1 | 34.9 | 147.0 | 24.3 |

# C Qualitative results

## C.1 Visualization

In this section, we provide the qualitative results of SageMix. As in Figure 1 and Figure 2, given original samples (left and right), SageMix generate the augmented samples (middle). Also, we qualitatively compare SageMix with other baselines in Figure 3.

# D Negative societal impacts and limitations

## D.1 Negative Societal Impacts

SageMix is designed for alleviating the problems of overfitting and data scarcity. To the best of our understanding, SageMix has no direct negative societal impact. However, similar to previous augmentation methods, our framework can be misused for malicious application. Especially, point clouds are widely used in various domains such as autonomous self-driving cars. In the real world, we cannot guarantee that virtual samples generated by data augmentation are always helpful for models to recognize objects. To mitigate this potential problem, we need additional verification for data augmentation methods.

---

[6]©2021 Jang-Hyun Kim, Wonho Choo, Hosan Jeong, and Hyun Oh Song. Licensed under MIT License

## D.2 Limitations

Since SageMix calculates point-wise weights using the RBF kernel, an additional hyperparameter $\sigma$ is required. Despite the consistent improvements, we empirically observed that the performance slightly varies according to the bandwidth. Although we demonstrated that our framework improves dense representation, as shown in part segmentation experiments, other localization tasks such as object detection have not been studied with our method. We believe that our method can be extended to diverse tasks including scene segmentation and object detection on indoor and outdoor scene point cloud datasets. These are left for future work.

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

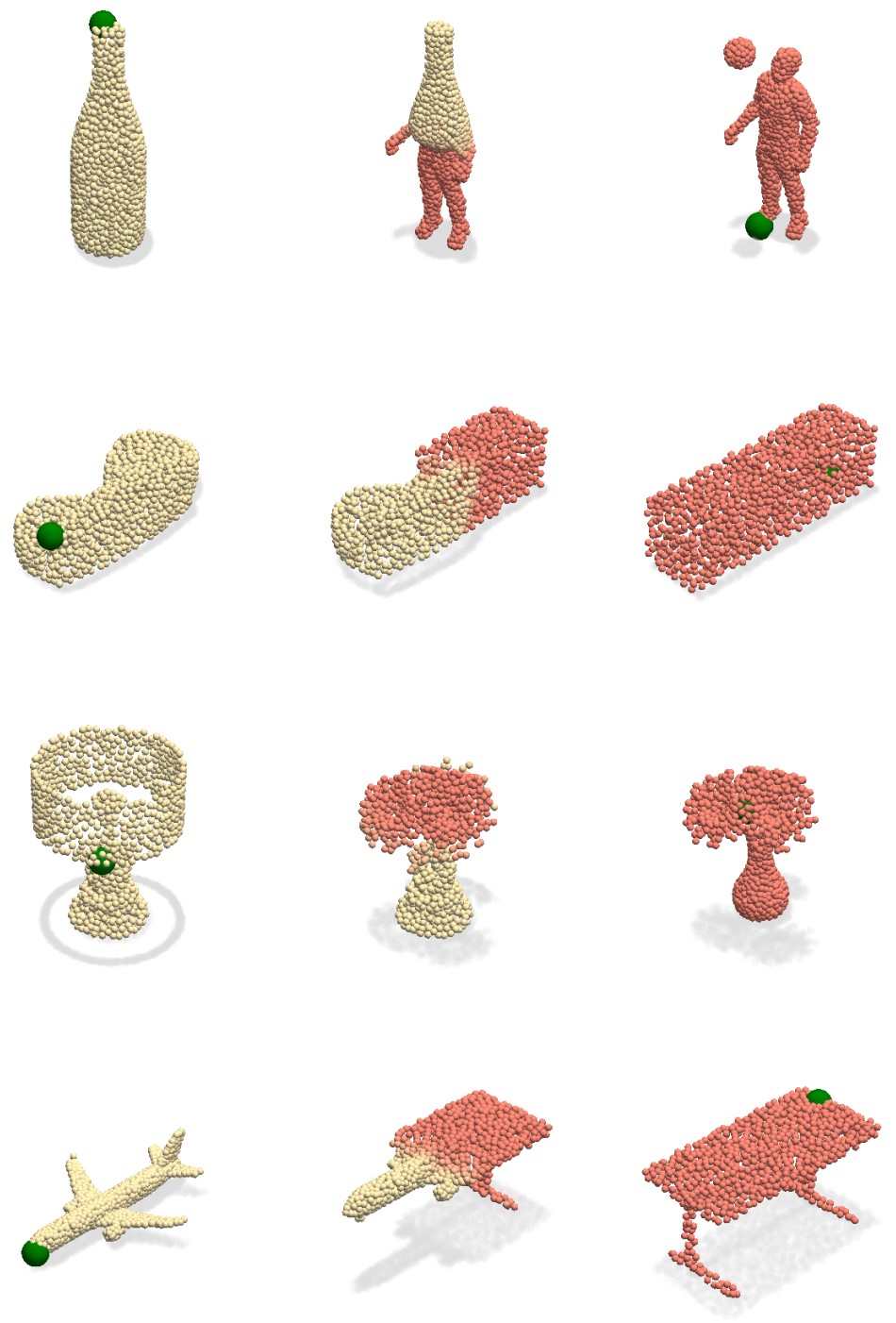

Figure 1: **Visualization of augmented samples by SageMix.** Given two samples (left and right), SageMix generates a sample (middle) based on query points.

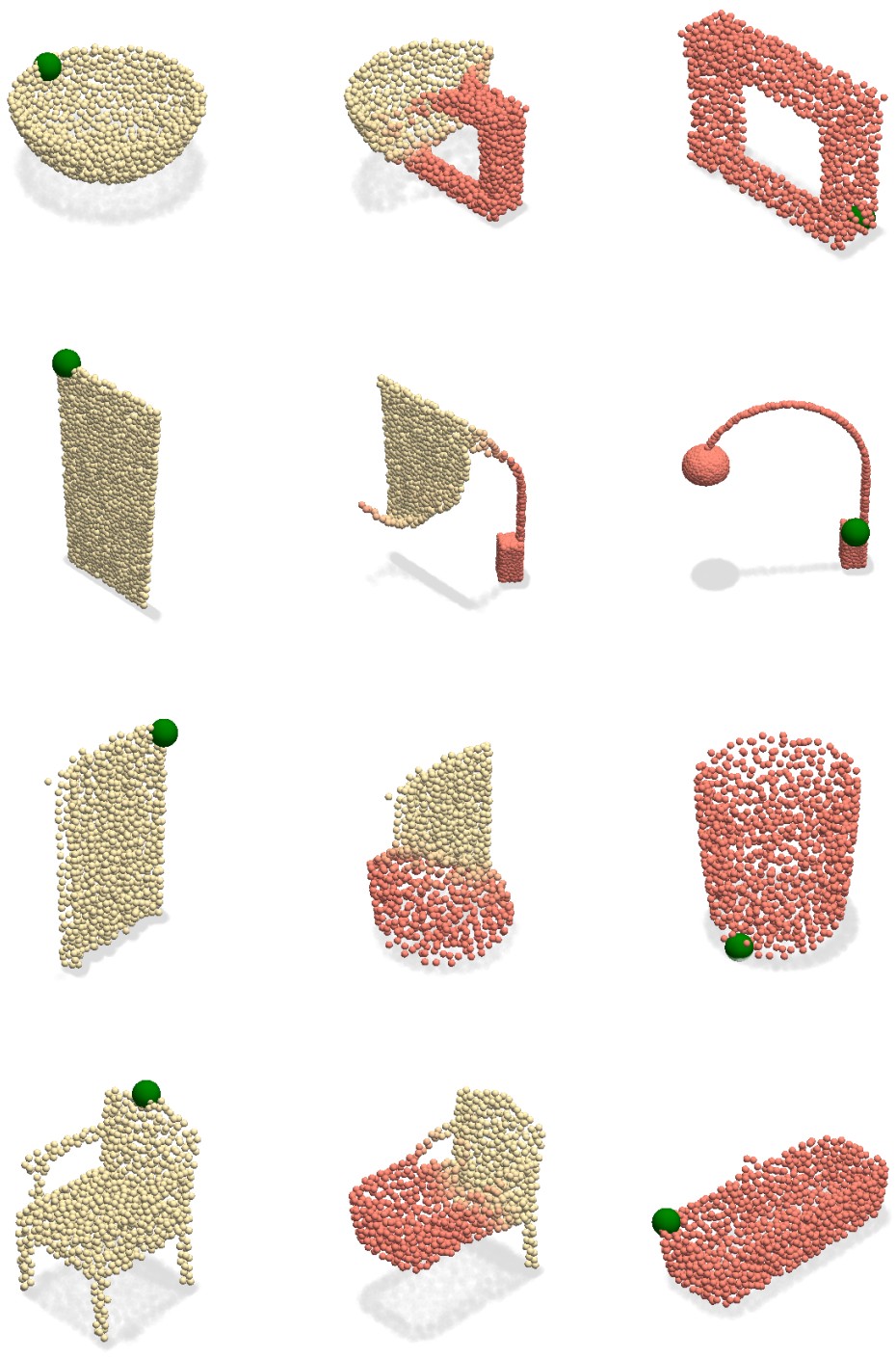

Figure 2: **Visualization of augmented samples by SageMix.** Given two samples (left and right), SageMix generates a sample (middle) based on query points.

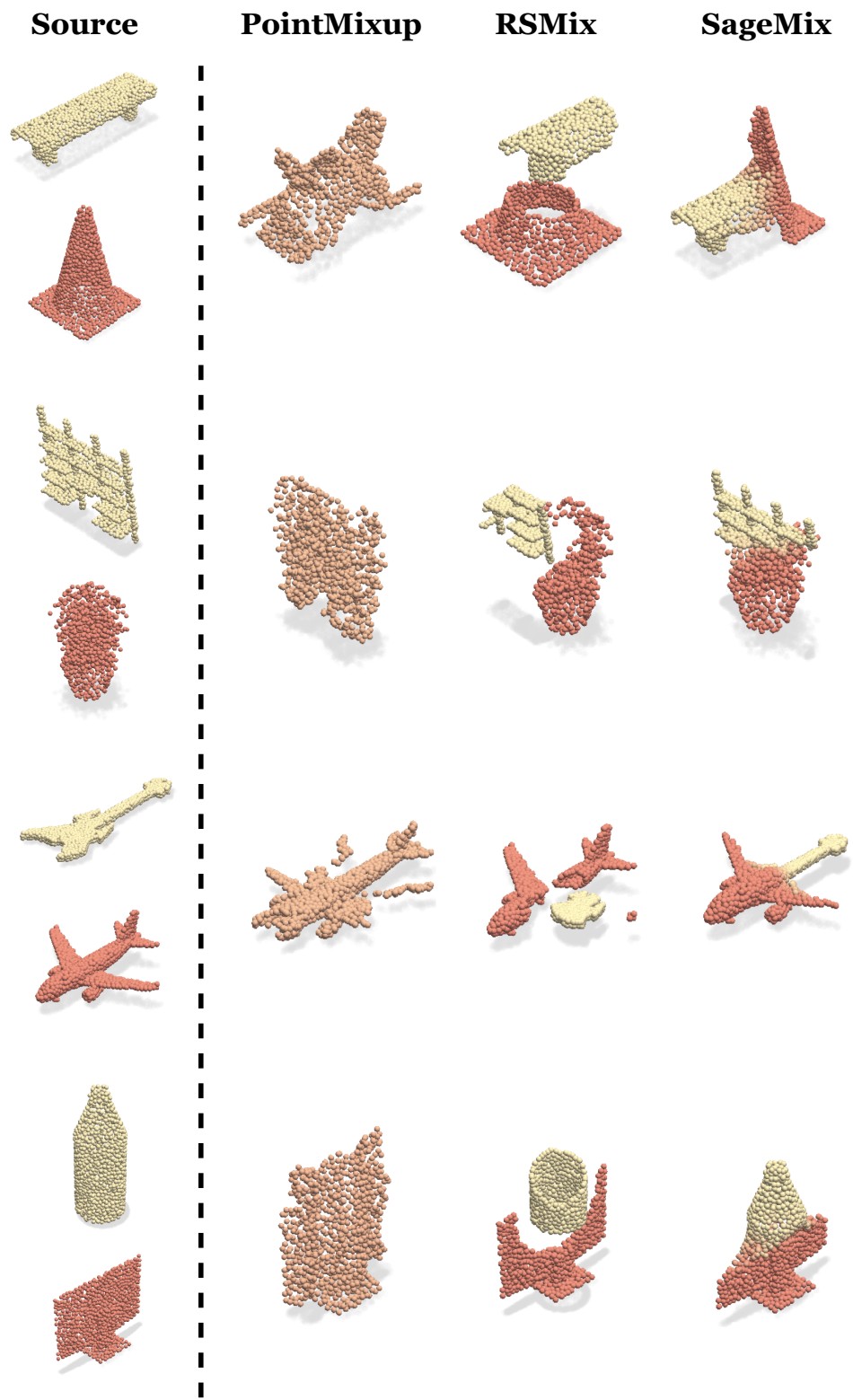

Figure 3: **Qualitative results with SageMix and baselines.** Given two source samples(left), PointMixup does not preserve the salient structure and RSMix loses the continuity. SageMix generates a continuous mixture preserving the local structure of original shapes(right).