# OpenReview forum: "SageMix: Saliency-Guided Mixup for Point Clouds"
_NeurIPS.cc/2022/Conference — NeurIPS 2022 Accept_

### Official Review · Reviewer_yU1N · 2022-07-10

**Rating:** 7
**Confidence:** 3
**Soundness:** 4 excellent
**Presentation:** 4 excellent
**Contribution:** 4 excellent

**Summary:**

This paper proposes a novel saliency guided mixup method for point clouds. It first utilizes saliency to find a query point for each of the two point clouds. Then it uses an RBF kernel (around the query point) to calculate the blending weights for each of the points. This method generalizes PointMixup [2], and shows superior performance against existing ones.

**Questions:**

- About the robustness experiments in Table3. Why don’t the authors present all tests presented in Table & in RSMix [15] ? e.g. Rotation 90, Rotation Y, Scale 1.4 are missing.
- The authors mentioned their varying performance with different $\sigma$ in Appendix D.2. However, they didn’t quantitatively experiment the performance except for a qualitative one in Figure 3. How does $\sigma$ affect the performance of SageMix?
- Another minor issue is missing reference of a paper on arxiv (non peer reviewed, but with decent citations). PointCutMix: Regularization Strategy for Point Cloud Classification. In ArXiv. https://arxiv.org/abs/2101.01461.

**Strengths And Weaknesses:**

Strengths
- The idea of introducing saliency guidance to the mixup method in point clouds sounds reasonable, as it has been proven to be useful in image-based methods [11] [12] [26].
- The design of sequentially sample two remotely located query (salient) points is delicate. It avoids the overlapping problem while preserving the important local structure of the two point clouds.
- The experiments seem sufficient.

Weaknesses
- I didn’t see any major weaknesses in this paper. One potential improvement to the method could be rotating and translating the whole point clouds to move the two query points far away from each other. In my opinion, it’s a more elegant way. Nevertheless, I think it’s fine to leave it as a future work.

---

> ### Author Response · Authors · 2022-08-01
> **Response to Reviewer yU1N**
>
> We appreciate the Reviewer yU1N for strong support to SageMix and detailed comments. We will address all of the concerns raised and incorporate them into the final version.
>
> ---
>
> **Comment 1:** One potential improvement to the method could be rotating and translating the whole point clouds to move the two query points far away from each other.
>
> **Answer:** Great question! Indeed, we have considered exactly the same technique as your suggestion. The rotation-translation method in most cases successfully preserves salient parts but we observed that it fails in some corner cases. We provide the visualization of **failure cases of the rotation-translation approach in Section C.2 of the revised appendix**. For instance, when the query points are located at the center of objects, then with any rotation and translation the augmented samples will lose the salient local structure. In SageMix, since the weight $w^t_i$ in **Equation (6)** for Mixup is computed based on the distance from each point $p^t_i$ to the query point $q^t$ in point cloud $\mathcal{P}^t$, rotating or translating a point cloud including the query points does not affect the weights (See the column $w^\alpha_i, w^\beta_i$ in Figure 5 of the appendix). Further, despite the various cases of rotation and translation, the points around a query point still correspond to the salient part in another point cloud (See the column $\phi$). As a result, the local structure of the salient parts is distorted in the augmented sample $\tilde{P}$.
>
> Our preliminary experiment shows that our saliency-guided sequential sampling is more suitable for SageMix. But, except for these extreme cases, Reviewer yU1N’s suggestion is also effective.
>
> **Equation (6)** : $w^t_i = K_\sigma(p^t_i, q^t) = \text{exp}\left(-\frac{\|p^t_i-q^t\|^2}{2\sigma^2} \right)$, where $t \in \{\alpha, \beta\}$
>
> ---
>
> **Comment 2:** Why don’t the authors present all robustness tests presented in RSMix. e.g. Rotation 90°, Rotation Y, Scale 1.4.
>
> **Answer**: Thanks for the detailed feedback on experimental settings. As suggested, we provide the additional experimental results below. As shown in the table, SageMix still achieves the best robustness on every corruption presented in RSMix.
>
> | Method | X-axis 90° | Y-axis 90° | Z-axis 90° | Y-axis 180° | scale 1.4 |
> | --- | --- | --- | --- | --- | --- |
> | Base | 11.3 | 86.0 | 13.3 | 86.1 | 82.1 |
> | + PointMixup | 12.4 | 86.2 | 13.9 | 86.2 | 82.2 |
> | + RSMix | 13.2 | 86.3 | 14.2 | 86.1 | 83.0 |
> | + **SageMix** | **15.2** | **87.1** | **14.7** | **87.2** | **84.7** |
>
> ---
>
> **Comment 3:** How does $\sigma$ affect the performance of SageMix?
>
> **Answer**: Here, we share the quantitative analysis of the bandwidth $\sigma$ with DGCNN and OBJ_ONLY. We observed that SageMix with a wide range of bandwidth $\sigma$ (0.1 to 2.0) consistently outperforms previous Mixup methods (e.g., 86.9%, 86.6% for PointMixup, RSMix). We will provide this analysis in the final version.
>
> |  $\sigma$ | 0.1 | 0.3 | 0.5 | 1.0 | 2.0 |
> | --- | --- | --- | --- | --- | --- |
> | OA | 87.2 | 88.0 | 87.6 | 87.3 | 87.6 |
>
> ---
>
> **Comment 4:** (Minor issue) Missing reference of a paper on arxiv (PointCutMix).
>
> **Answer**: PointCuxMix is non-peer-reviewed, so we did not include it in our submission. However, we agree with Reviewer yU1N, and we will include it in the final version.

---

> > ### Author Response · Authors · 2022-08-08
> > **A Reminder of the Author-Reviewer Discussion**
> >
> > Dear Reviewer yU1N, we appreciate the reviewer for constructive feedback and comments.
> >
> > The end of the Author-Reviewer Discussion is close. Through rebuttal, we have addressed all your concerns, and we believe that our responses have answered your suggestions and questions. So, would it be possible to check our responses and let us know if you have any concerns or questions unresolved?
> >
> > Once again, we appreciate your efforts in reviewing our paper.
> >
> > Sincerely, Authors

---

### Official Review · Reviewer_YgrL · 2022-07-11

**Rating:** 4
**Confidence:** 4
**Soundness:** 2 fair
**Presentation:** 3 good
**Contribution:** 2 fair

**Summary:**

This paper presents a method for the data augmentation of 3D point clouds. The proposed Saliency-Guided Mixup for point clouds (SageMix) preserves discriminative local structures and generates continuous samples with smoothly varying mixing ratios. Here saliency is computed based on the impact to the corresponding task, measured through the gradients of the loss. Experimental results show that SageMix brings consistent and significant improvements over state-of-the-art Mixup methods.

**Questions:**

1. Why 3D mixup is more challenging than 2D mixup? What is the particular difficulty of extending mixup from 2D to 3D?
2. The computation of saliency is similar to the popular CAM approach. What is the impact of using different methods for saliency compuation?
3. I suggest the authors make a more solid evaluation for the proposed method.

**Strengths And Weaknesses:**

Strengths:
1. The saliency-guided sequential sampling is technically novel.
2. There are some ablation studies to demonstrate the effect of the proposed method.
3. Overall, the paper is well organised.

Weaknesses:
1. The difference between 2D and 3D mixup-based methods is not insightfully analysed in the introduction.
2. The shape-preserving continuous mixup component just follows the mainstream method and thus the novelty is limited.
3. The experimental results is not extensive. The proposed method is only quantitatively compared with PointMixup and RSMix. And there is also a lack of qualitative results.

---

> ### Author Response · Authors · 2022-08-01
> **Response to Reviewer YgrL (2/2)**
>
> **Comment 4:** The computation of saliency is similar to the popular CAM approach. What is the **impact** of using different methods for saliency computation?
>
> **Answer:** Great question! There are several ways to calculate a saliency map in the 2D image domain [1-5]. However, the methods for 2D images cannot be directly applicable to the point cloud domain due to the lack of background, especially in single-object point cloud data. Most methods in the 2D image domain focus on detecting salient "foreground objects" whereas in our setting we need to detect salient parts of a single object. In addition, [3-5] utilize the additional network to detect salient regions in the supervised setting with the ground truth of foreground objects.  Since no ground truth label for saliency detection is available and we wanted to minimize computational overhead for saliency detection, we simply used the norm of gradient as PuzzleMix [6] and CoMixup [7]. Also, in a preliminary experiment, we observed that SageMix achieves slightly higher performance with this simple gradient-based saliency map than PointCloud Saliency Maps [8].
>
> [1] Li et al. “Robust Saliency Detection via Regularized Random Walks Ranking” Proceedings of the IEEE conference on computer vision and pattern recognition, 2015
>
> [2] Zhu et al. “Saliency Optimization from Robust Background Detection”, CVPR 2014
>
> [3] Deng et al. “3Net: Recurrent Residual Refinement Network for Saliency Detection”, IJCAI 2018
>
> [4] Liu et al. “PiCANet: Learning Pixel-wise Contextual Attention for Saliency Detection”, CVPR 2018
>
> [5] Qin et al. “BASNet: Boundary-Aware Salient Object Detection”, CVPR 2019
>
> [6] Kim et al. "Puzzle mix: Exploiting saliency and local statistics for optimal mixup", ICML 2020
>
> [7] Kim et al. "Co-Mixup: Saliency Guided Joint Mixup with Supermodular Diversity", ICLR 2021
>
> [8] Zheng et al. “PointCloud Saliency Maps”, ICCV 2019

---

> > ### Author Response · Authors · 2022-08-08
> > **A Reminder of the Author-Reviewer Discussion**
> >
> > Dear Reviewer YgrL, we appreciate the reviewer for constructive feedback and comments.
> >
> > The end of the Author-Reviewer Discussion is close. Through rebuttal, we have addressed all your concerns, and we believe that our responses have answered your suggestions and questions. So, would it be possible to check our responses and let us know if you have any concerns or questions unresolved?
> >
> > Once again, we appreciate your efforts in reviewing our paper.
> >
> > Sincerely, Authors

---

> ### Author Response · Authors · 2022-08-01
> **Response to Reviewer YgrL (1/2)**
>
> **We appreciate the Reviewer YgrL for the acknowledgment of the novelty of our saliency-guided sequential sampling. We will address all of the concerns raised and incorporate them into the final version.**
>
> ---
>
> **Comment 1:** The difference between 2D and 3D mixup-based methods is not insightfully analyzed in the introduction. Why 3D mixup is more challenging than 2D mixup? What is the particular difficulty of extending Mixup from 2D to 3D?
>
> **Answer:** We apologize for the brief explanation of the difference between 2D and 3D Mixup. As we mentioned in line 29-30 of the main paper, one of the main differences in point clouds is that the data has an **unordered and non-grid structure**. 2D images on a regular grid space naturally come with one-to-one correspondence between pixels. So, 2D mixup can be defined by a simple interpolation of pixel values.  However, in the point cloud domain, there is **no one-to-one correspondence between two point clouds**. In addition, unlike 2D images with RGB values at each pixel, **3D point clouds have no feature** at each point except for the coordinates. Hence, even if the correspondence between two point clouds is estimated, naive interpolation of the coordinates (e.g., PointMixup) will destroy important local structures of the original point clouds. For these reasons, it is challenging to devise effective 3D Mixup for point clouds by simply adopting 2D Mixup methods.
>
> ---
>
> **Comment 2:** **The shape-preserving continuous mixup component just follows the mainstream** method and thus the novelty is limited.
>
> **Answer:** We are glad that Reviewer YgrL agrees on the importance of shape/continuity preserving in 3D Mixup. But we want to point out that **developing a method in a right direction does NOT limit the novelty**. More importantly, no previous method in 3D Mixup achieved both shape-preservation and continuity of augmented samples. For instance, PointMixup generates continuous samples but it causes a huge distortion of local structures. On the other hand, RSMix, which is an adoption of CutMix in 3D, preserves the local structure of an extracted region but the resulting samples are discontinuous. To the best of our knowledge, **SageMix is the first work, in the point cloud domain, that combines two point clouds into one continuous shape while preserving the local structure.** Further, as Reviewer YgrL mentioned, based on **novel saliency-guided sequential sampling**, SageMix preserves the shape of the salient region. Our ablation study shows that it further improves generalization ability. Note that no previous 3D Mixup utilized saliency information. Considering these aspects, our contributions are significant.
>
> ---
>
> **Comment 3-1:** The experimental results are not extensive. I suggest the authors make a more solid evaluation for the proposed method.
>
> **Answer:**  In the main paper, **we provided experimental results on 4 benchmark datasets (ModelNet40, ScanObjectNN, ShapeNetPart, and CIFAR-100) in 5 different tasks**: 2D/3D classification, part segmentation, uncertainty calibration, and robustness evaluation. Since 3D Mixup has been relatively less explored, only recent works PointMixup (2020) and RSMix (2021) are included as our baseline methods. Following PointMixup and RSMix, we provided experimental results only on ModelNet40. To provide more solid evaluation, we additionally conducted experiments in classification with two splits (OBJ_ONLY, and PB_T50_RS) of ScanObjectNN. In addition, we reported uncertainty calibration errors that are not studied by previous 3D Mixup methods and we evaluate the effectiveness of SageMix in part segmentation using ShapeNetPart. Finally, SageMix was applied to 2D images (CIFAR-100) with minor modifications. If any specific experiments are needed, please let us know. We’re willing to provide more experimental results.
>
> **Comment 3-2:** The proposed method is only quantitatively compared with PointMixup and RSMix. And there is also a lack of qualitative results.
>
> **Answer:**  In the main paper, we provide various qualitative results in the main paper/supplement. Figure 1 visually compares our method with baselines: PointMixup and RSMix. In addition, we provided sensitivity analysis of hyperparameters (prior factor $\pi$ and bandwidth $\sigma$) in our method in Figure 3. Augmented samples by our method are presented in Figure 1-3 of the appendix. Also, considering Reviewer YgrL’s suggestion, we include more qualitative results to compare augmented samples by ours and baselines (**Figure 4 of the revised appendix**).

---

### Official Review · Reviewer_VLSt · 2022-07-12

**Rating:** 5
**Confidence:** 4
**Soundness:** 3 good
**Presentation:** 3 good
**Contribution:** 2 fair

**Summary:**

The paper proposes SageMix, a data augmentation technique for point clouds. Similar to the Mixup family of data augmentations, SageMix mixes two point clouds. It tries to mix point clouds in a saliency-guided way to preserve the salient local structures. Experiments have been conducted to show the efficacy of the method.

**Questions:**

Refer to the weakness section for questions. Overall, I am ambivalent about the paper. The method could be useful but I am unable to conclude from the experiments because of reasons mentioned in the weakness sections. Hence, I suggest a borderline rating for now. I will update the score based on the rebuttal.

**Strengths And Weaknesses:**

Strengths:

- The paper is well-written and easy to follow.
- It is nice that this data augmentation can lead to more robust networks as shown in Table 3.

Weakness:

- A critical weakness is that experimental findings do not appear to be very significant. The performance difference between various techniques is not very large and no error margin has been reported (Table 2). It would be nice if the paper could provide some mean +- std measures. This could be done by running the same experiments multiple times (with random initialization) and reporting the mean and variance. This is particularly important as point-based benchmark methods can have significant variations across runs.

- Most experiments in the paper are limited to point cloud classification. The experiment on part-segmentation has not been described in detail. It would be nice to see experiments similar to Table 2 for segmentation as well. This would help in showing that the technique can be used beyond classification.

---

> ### Author Response · Authors · 2022-08-01
> **Response to Reviewer VLSt**
>
> **We appreciate the Reviewer VLSt for supportive comments and constructive feedback on our work. We will address all of the concerns raised and incorporate them into the final version.**
>
> ---
>
> **Comment 1-1**: A critical weakness is that experimental findings do not appear to be very significant. The performance difference between various techniques is not very large.
>
> **Answer:** Compared to the state-of-the-art Mixup methods, the improvement by SageMix are promising. Our SageMix with PointNet++ achieved **2.6%, 1.7%, 4.0%** improvements over a standard training in ModelNet40, OBJ_ONLY, and PB_T50_RS, respectively. The performance gap over the second-best techniques are **1.0%, 0.6%, and 2.6%**, which is significant. We also observed similar improvements with DGCNN by **1.1%(OBJ_ONLY), 1.1%(PB_T50_RS)** over previous SOTA methods. Lastly, the performance gain with PointNet seems relatively small (e.g., +0.4%(ModelNet40), +0.1%(OBJ_ONLY), +0.4%(PB_T50_RS)) compared to the second best techniques but we believe that this is mainly due to **the limited capacity of PointNet**, which is a nascent model for point clouds only with MLPs. Our experimental results evidence that our method significantly boosts performance as long as the model has sufficient capacity.
>
>
> **Comment 1-2:** It would be nice if the paper could provide some mean +- std measures.
>
> **Answer:** Great Point! As Reviewer VLSt mentioned, performance oscillation is an important issue in the point cloud benchmarks. However, for a fair comparison with the numbers reported in PointMixup and RSMix, we followed the prevalent evaluation metric in point clouds, which reports the best validation accuracy. Apart from this, we here provide the additional results with five runs on OBJ_ONLY. The mean and standard deviation are presented in the table below.
>
> | Method | PointNet | PointNet++ | DGCNN |
> | --- | --- | --- | --- |
> | Base | 78.56±0.51 | 86.14±0.39 | 85.72±0.44 |
> | +PointMixup | 78.88±0.28 | 87.50±0.26 | 86.26±0.34 |
> | +RSMix | 77.6±0.56 | 87.30±0.65 | 85.88±0.59 |
> | +**SageMix** | **79.14±0.30** | **88.42±0.26** | **87.32±0.53** |
>
> It is worth noting that SageMix consistently achieves the best performance with **significant improvements over the second-best methods**. These improvements prove the effectiveness of SageMix. We will provide this result in the appendix.
>
> ---
>
> **Comments 2:** It would be nice to see experiments similar to Table 2 for segmentation as well.
>
> **Answer:** Thanks for Reviewer VLSt’s constructive suggestions. Part segmentation is one of the major tasks in point cloud processing. However, PointMixup and RSMix did not demonstrate their methods in part segmentation and no number was reported for ShapeNetPart. So, we reported the performance of our method only in the main paper. In addition, we provided the detailed results of part segmentation with and without SageMix in Table 3 of the appendix. As suggested, we compare our method with PointMixup and RSMix for part segmentation. We used the official code by the authors with minor modifications for generating point-wise ground truth. The results are summarized in the table below.
>
> |  Method | DGCNN | PointNet++ |
> | --- | --- | --- |
> | Base | 85.1 | 85.1 |
> | +PointMixup | 85.3 | 85.5 |
> | +RSMix | 85.2 | 85.4 |
> | +**SageMix** | **85.4** | **85.7** |
>
> Note that although the gain seems small, SageMix outperforms previous Mixup methods. Also, considering the already saturated performance of ShapeNetPart, we believe that the improvement (+0.3%, +0.6% in DGCNN, PointNet++) over the base model is not trivial. We will reflect this table in the final version as well.

---

> > ### Author Response · Authors · 2022-08-08
> > **A Reminder of the Author-Reviewer Discussion**
> >
> > Dear Reviewer VLSt, we appreciate the reviewer for constructive feedback and comments.
> >
> > The end of the Author-Reviewer Discussion is close. Through rebuttal, we have addressed all your concerns, and we believe that our responses have answered your suggestions and questions. So, would it be possible to check our responses and let us know if you have any concerns or questions unresolved?
> >
> > Once again, we appreciate your efforts in reviewing our paper.
> >
> > Sincerely, Authors

---

### Meta-Review · Area_Chair_W8mD · 2022-08-26

**Recommendation:** Accept
**Confidence:** Certain

**Metareview:**


This paper studies the point cloud data mixup with the saliency guidance. The proposed SageMix focus on the mixup over the local regions to preserve salient structures which are more informative for downstream tasks. The whole paper is well organized with clear logic to follow. The proposed method is simple but effective. Moreover, there are solid experiments in various tasks, including object classification, parts segmentation and calibration, to comprehensively evaluate proposed methods. One of the major concerns is the limited improvements over the standard mixup (Reviewer VLSt) on PointNet++. And the discussion of 2D and 3D mixup can be enriched in the aspects of technical challenges and novelties (Reviewer YgrL). This paper includes five different tasks and four benchmarks in experimental studies that strongly address the third major concern in the limited evaluation of Reviewer YgrL, who, however, has not provided any feedback after the authors' rebuttal. Considering the overall contributions in methods and solid evaluation, this submission is slightly above the bar of acceptance.

**Award:**

No

---

### Decision · Program_Chairs · 2022-09-14

Accept